# Polymeric Materials and Microfabrication Techniques for Liquid Filtration Membranes

**DOI:** 10.3390/polym14194059

**Published:** 2022-09-27

**Authors:** Thomas Kerr-Phillips, Benjamin Schon, David Barker

**Affiliations:** 1School of Chemical Sciences, University of Auckland, Private Bag, Auckland 92019, New Zealand; 2The MacDiarmid Institute for Advanced Materials and Nanotechnology, Victoria University of Wellington, Wellington 6012, New Zealand; 3The New Zealand Institute for Plant and Food Research Limited, Canterbury Agriculture and Science Centre, 74 Gerald St., Lincoln 7608, New Zealand

**Keywords:** liquid filtration, polymers, membranes

## Abstract

This review surveys and summarizes the materials and methods used to make liquid filtration membranes. Examples of each method including phase inversion, electrospinning, interfacial polymerization, thin film composites, stretching, lithography and templating techniques, are given and the pros and cons of each method are discussed. Trends of recent literature are also discussed and their potential direction is deliberated. Furthermore, the polymeric materials used in the fabrication process of liquid filtration membranes are also reviewed and trends and similarities are shown and discussed. Thin film composites and selective filtration applications appear to be a growing area of research for membrane technology. Other than the required mechanical properties (tensile strength, toughness and chemical and thermal stability), it becomes apparent that polymer solubility and hydropathy are key factors in determining their applicability for use as a membrane material.

## 1. Introduction

Filtration is a processing aid that is utilized across a wide range of industries, with its value growing rapidly and expected to reach USD 2.9 billion by 2024 [1]. It is a key technology within the wine, dairy and water purification industries, with many more esoteric applications, such as membranes developed for research purposes, selective removal or capture of targeted species [2,3,4,5,6,7,8]. This can be attributed to the many benefits filtration techniques have when compared to other purification techniques, which include the relatively low energy cost, the simplicity of setup, its non-destructive nature (keeping important components in the liquid intact), little to no waste and it does not require the addition of any external additives. Liquid filtration is a very well-established technology, with early water filtration systems dating back to around 500 BC with the Hippocratic Sleeve. Water filtration first became industrialized in the 1800s, beginning with the use of sand filters to control the spread of cholera [9]. However, it wasn’t until 1937, with the invention of nylon, that saw the first synthetic polymers being used a membrane material [10] and even then, bio-based polymers such as cellulose acetate were still the dominant material [10]. Research in membrane technology is currently focused on mechanically stable, more efficient filters with better rejection, allowing membranes to last longer with lower energy costs. Today, water purification and beverage clarification are among the most common users of liquid filtration, each with their own unique filtration requirements.

The objective of a liquid filtration membrane is to separate particulates from a liquid source, producing a purified output stream and leaving behind a concentrated source. Liquid filtration is categorized based on the size of particles rejected or passed (Figure 1). Membrane performance is evaluated based on transmembrane flux rates, retention/rejection efficiency and long-term stability. One of the main limitations of filters is their rapid fouling [11,12,13,14,15,16], thus a large focus of recent research is aimed at fabricating filters with anti-fouling properties, improving their long-term stability. A related focus is on fabricating filter membranes with specific rejection/retention, which will allow the removal or concentration of undesirables/desirables.

In this review we summarize and discuss polymeric materials and the fabrication methods used to produce polymer membranes for liquid filtration, as well as methods used to modify these filters to enhance their performance. While reverse osmosis (RO) membranes will be mentioned, they will not be discussed in detail as they work on a fundamentally different mechanism to other filtration membranes. However, a detailed discussion on RO membranes can be found in the works by Rana et al. [17] and another detailing specifically thin film composite RO membranes by Ng et al. [18]. Additionally, absorbent materials such as metal organic frameworks (MOF)s [19], ceramics [20] and activated carbon [21], will not be discussed as this review will focus on filters fabricated from polymeric materials.

## 2. Microfabrication Techniques

Microfabrication is the process by which a device is made with at least one dimension a micrometer or smaller. By this definition almost all modern, and some traditional, membrane fabrication techniques are microfabrication. A successful fabrication process will create a membrane with the highest porosity (for the targeted filtration level), of sufficient mechanical strength and is scalable to fabricate membranes of adequate size. Herein we will discuss the microfabrication methods used by modern processes and researchers, their advantages, disadvantages and their future directions. These will include: phase inversion, electrospinning, interfacial polymerization, stretching, templating, lithography and self-assembly. It is worthy to note that the majority of commercially available filter membranes are prepared by the phase inversion technique (see Synder XT membranes as an example), with some older products still being fabricated using the stretching method (some Whatman and Millipore products as examples).

### 2.1. Phase Inversion

Phase inversion has been utilized to fabricate porous substrates for several decades [22] and is still today one of the more common methods for the fabrication of filters. There are four different types of phase inversion techniques: precipitation from the vapor phase, precipitation by controlled evaporation, thermally induced phase separation and immersion precipitation. A detailed description of these techniques can be found in a review by Holda et al. [22] but are briefly outlined here. Precipitation from the vapor phase involves exposing a polymer solution to a non-solvent gas saturated with the solvent the polymer is dissolved in. The good solvent (which the polymer is dissolved in) in the gas prevents this solvent from evaporating and instead the non-solvent diffuses into the polymer solution, resulting in precipitation and formation of porous features. Precipitation by controlled evaporation is performed using the polymer dissolved in a solvent and non-solvent mixture, where the solvent is considerably more volatile, this results in the solvent evaporating at a greater rate, thus allowing the polymer to precipitate out, forming a porous film. Thermally induced phase separation involves reducing the temperature of a polymer solution, triggering phase separation, a membrane is formed upon solvent evaporation. Lastly, immersion precipitation involves loading a polymer solution onto an appropriate support that is then pulled through a coagulation bath which contains a non-solvent, causing a porous film to form on contact. 

The main advantages of this technique are outlined below. It is a scalable process, as there are no scale limiting steps in this process and thus it can be tailored to produce membranes of any size. This method can naturally form something similar to a thin film composites, with a thinner smaller dimensioned pore layer and wider, more porous support layer (Figure 2). This is called the Loeb-Sourirajan structure and has been shown to have several advantages [17,23,24]. Phase inversion also has a wide selection of polymer materials available to it which is both the cause and a result of it prolific use.

The main disadvantages of this technique include the relatively complex apparatus required to control the vapor phase environment and temperature. Furthermore, the required solvents are often hazardous (such as dimethylformamide or chloroform). Lastly, with the expectation of immersion precipitation, it is difficult to produce a continuous manufacturing process.

### 2.2. Electrospinning

Of the filter fabrication techniques being researched in recent years, electrospinning is one of the more common techniques employed. This is likely due to its simplicity and capability of large-scale production [26]. Electrospinning is not a new invention [27], but has gained momentum due to the recent drive in nanotechnology. As a result, the technique has seen a growing number of industrial applications as well as a growing number of polymer materials available to it.

The technique involves preparing a viscoelastic polymer solution which is delivered through a spinneret tip. A large electric potential difference is applied across the spinneret and a collector (which is usually earthed). As the electric field overcomes the solution surface tension, a Taylor cone [27] forms at the needle tip and a solution jet will escape the droplet and fly towards the collector. During flight, a whipping instability causes the jet to whip erratically, stretching the jet, causing it to become thinner, and allowing solvent evaporation to occur. In this manner, mats of randomly-oriented fibers with widths in the nano-to mico- range are deposited on the collector (Figure 3). 

There are several advantages of this technique compared to other microfabrication methods for filtration membranes, these include the following. The large surface to volume ratio of an electrospun membrane and its high porosity [26,29,30] can lead to enhanced flux rates and improved efficiency compared to conventional fabrication techniques [31]. Its simple apparatus, ease of setup and production, making it relatively straightforward for researchers to build or purchase a lab scale system and allowing for industrial scalability. There exists a wide range of polymer materials capable of electrospinning and this is still growing. Additionally, other materials can be included in the electrospinning solution, with a polymer in solution as a carrier [32,33]. This can be important for fabricating anti-fouling or selective filter membranes. Another advantage is its continuous nature. Electrospinning produces a continuous fiber and can be adapted to a continuous sheet production, making it perfect for a continuous industrial fabrication process. There are many easily adjustable parameters that govern the production of an electrospun fiber membrane. This makes electrospinning a very tunable process; porosity [26,29,30], fiber diameter [27,28,29,34], fiber shape [34,35,36] and fiber orientation [37,38,39] can all be controlled during electrospinning by adjusting the some of these parameters.

However, like all methods, there are disadvantages. Like phase inversion, electrospining often requires hazardous solvents. While there are some relatively non-hazardous solvents available (such as water, formic acid or methyl ethyl ketone [40,41]), the majority of solvents used are often flammable or toxic (such as dimethylformamide, chloroform or tetrahydrofuran [15,42,43]). Another disadvantage is that there is usually some material loss which occurs with most electrospinning setups. Due to the uncontrolled nature of the whipping instability, some material will fall outside the region of the collector. This can be taken into account when setting up the apparatus however, reducing this drawback. Additionally, due to the inherent random fiber deposition, a wide range of pore sizes are produced. This is only a drawback for situations where a particular size rejection is desired and extensive research has been conducted in this area to diminish this drawback [38,44,45,46]. Another drawback to electrospinning is that it can be sensitive to ambient relative humidity. The electrospinning process utilizes an electric field between to a highly charged needle and a grounded plate. Sparking occurring in an often-flammable environment is an obvious fire hazard. Therefore, the air resistance between the two is a factor that has to be considered. Often, if the humidity is too high, an electric field strong enough, without sparking between the collector and needle, cannot be achieved to commence electrospinning. This can be avoided with adequate humidity control but not all electrospinning apparatus has such control available.

### 2.3. Interfacial Polymerization and Thin Film Composites

Interfacial polymerization is a secondary membrane fabrication technique. Here, a pre-existing porous membrane is swollen with a solvent and polymer precursor, the swollen membrane is then exposed to a solution that is non-miscible with the first solvent and contains the remaining polymer precursor, such that a polymer forms at the solution-membrane interface [47]. This technique then allows for the creation of ultrafiltration or reverse osmosis membranes form micro- or nano- filtration membranes [18,48,49]. An added benefit of this method is that it can alter the hydrophilicity of the membrane surface, allowing for tailored compatibility with the desired filtrant solution [18,47,48,50]. As this is a secondary fabrication technique, its main drawback are the additional steps required over and above the steps to fabricate the supporting membrane.

This method aligns well with a recent trend within filtration membrane research, the development of thin film composites (TFCs) [18,47,49]. Typically, a TFC is made from a highly porous membrane with an extra barrier layer added, that provides nano, ultra or reverse osmosis (RO) filtration [18,47,48,49,50]. This makes interfacial polymerization receive additional attention as it is a primary method for forming TFCs. The benefit of a TFC is the added support from the underlying porous membrane, providing a mechanical strength that the thin barrier layer cannot. Furthermore, higher fluxes are possible, due to the porosity of the support layer and thinness of the flux-limiting barrier layer [18,47,50]. Such a membrane is fabricated with a technique such as (but not limited to) interfacial polymerization.

Notice how the TFCs structure in Figure 4 has similarities to the intrinsic structures formed in Figure 2 second column (e.g., d2), with a thick, highly porous layer and a thin, much denser, smaller pored layer, the Loeb-Sourirajan structure. This explains why phase inversion is such a popular choice for membrane fabrication as it can form this favorable structure without additional steps. However, the two layered system (as in Figure 4) is often superior to the coincidental one, as more control is possible over the membrane fabrication process as it is possible to specify the properties of both layers in a TFC [47].

### 2.4. Stretching

Like electrospinning and phase inversion, the stretching method is also a well-established technique. This process involves stretching a polymer film at low temperature to induce nucleation points followed by stretching at elevated temperatures to then cause a microporous structure to evolve (Figure 5) [51,52]. This technique requires highly crystalline polymers, with polypropylene being the most common and has an advantage of being solvent free [51,52]. However, relatively little research has been conducted on this technique over the last decade. This is likely due the small number of available polymer materials suitable and then their comparatively inferior properties (solvent resistance, temperature resistance, hydropathy, ability to functionalize etc.) compared to other polymers used in membranes. However, this method is still used industrially (see 3M™ Membrana™ Oxygenation Membrane Series).

### 2.5. Templating, Ablation, Photolithography, Etching and Self-Assembly

These techniques are all labor-intensive methods that are difficult to utilize on an industrial scale. Furthermore, when fabricating membranes using these techniques, often a combination of two or more of these techniques are used to form the resulting membrane. These techniques can be described as follows.

Photo-lithography: Photo-lithography is where a laser or light source is used with a mask to create a micro/nano patterned material. Etching and stripping is followed, resulting in a material with a precisely controlled pattern [54].

Ablation: Ablation is the process whereby part of a material is obliterated, by some means, in a controlled manner [55].

Templating: Templating involves using a, typically, nanoscale “mold” to fabricate porous polymer membranes [56,57]. This technique is often coupled with photolithography in the creation of the mold [56].

Self-assembly: Self-assembly takes advantage of a polymer system’s intrinsic nature to self-assemble into patterned phases. This is usually done with block polymers whose blocks comprise polymers that do not blend [58]. If the casting conditions are controlled, such polymers phase separate into distinct patterns. When using self-assembly to form filter membranes, usually one phase is water permeably or one phase can later be removed by other means, forming filtration membranes with very defined nanoscaled patterns [59]. A drawback to such a method is that it can only be used to make very thin membranes and is hard to upscale, however, like interfacial polymerization, this technique can be combined with others to form composite membranes.

Techniques such as templating, photolithography, etching and self-assembly techniques allow for much more control over intricate details and morphology [60,61,62] of the resulting membrane compared to the aforementioned fabrication techniques and are sometimes used sequentially [62]. These techniques are usually labor intensive, difficult to produce on the large scale and cannot be performed in a continuous process like electrospinning or immersion precipitation. As such, they have not found their way into the industrial membrane fabrication market, however, due to the morphological control they provide, these methods are ideal for more fundamental membrane research, linking morphology to performance [54]. The main advantage of these techniques is the precise control of the nano- and micro- morphology of the resulting membrane, making it an ideal method for selective filtration as the exact pore structure to reject specific particles of specific shape and size can be achieved [55,56,60,61,62].

A good example of these techniques (laser ablation) being used to study the effects of precise membrane morphology is provided by Alderson et al. [55]. In their study, they used laser ablation to generate an auxetic (a material with a negative Poisson ratio) polymer film. The pores in this material opened when stretched (Figure 6), allowing for an easy, mechanical antifouling process.

One of the benefits of utilizing a mold is that they do not always require a polymer solution. By removing the need of a solvent, the process becomes far less hazardous. As an example Fan et al. [56] created an alumina mold through lithography that then allowed polyethylene films to be hot pressed and imprinted by the mold (Figure 7), producing very defined, porous structure, with wine-bottle shaped pores. Their study revealed that this morphology had superior filtering properties compared to commercial filters with randomized pore structures.

## 3. Polymer Materials

To make an effective liquid filter, the material used must possess properties favorable for membrane fabrication such as ease of processing, amenability to clean-in-place procedures (CIP), mechanical robustness, appropriate hydropathy depending on the application, temperature-resistance, chemical inertness, cost-competitivity, and may need to be food-safe [24,51]. Materials which inherently have (or can be modified to provide) added functionality such as anti-fouling, or targeted filtration present increased attractiveness for use [24,51]. These are challenging criteria for any one material to fulfil, and most materials (including the polymer membranes commonly used by industry) are only able to fulfil some of these criteria.

A defining feature of polymeric materials is their ease of processing. Thus, there are many methods for creating highly porous membranes with appropriate pore sizes for the different levels of filtration. Furthermore, many polymers are easily chemically modified to allow for the creating of anti-fouling membranes and other more advanced filtration applications [25,63,64]. However, a critical factor in choosing a polymer material is the fabrication technique chosen to generate the membrane porosity, this is largely due to most techniques depending on the polymer solubility [22,51,65,66]. The dependence of the fabrication technique and polymer choice on solubility presents a unique challenge, as a desired characteristic for filter membranes is solvent resistance, however the fabrication techniques favors polymers that are readily soluble. This requirement of both requiring solubility for fabrication and possessing solvent resistance greatly narrows the range of applicable polymers.

Hydropathy is worth special mention. It has been shown that for aqueous systems, increasing membrane hydrophilicity will improve its anti-fouling properties [34,67,68]. With the theory that many fouling contaminants are hydrophobic and thus are repelled by a hydrophilic surface [34,67,69]. A polymer’s hydropathy or at least its ability to modify the surface functional groups then becomes important for membrane performance and thus for the decision of material choice.

A table expressing the physical properties of some of the common polymers used to make filters is shown in Table 1.

### 3.1. Polyethersulfone and Polysulfone

Of the common polymers used for aqueous filtration applications, polyethersulfone (PES) and polysulfone (PSf) are two of the more common, with many commercially available membranes [86]. This is largely due to their high strength and creep, temperature and chemical resistance [48,87,88]. However, one of the drawbacks with PES and PSf is their hydrophobicity; this likely is why recent research on PES/PSf filters focus on polymer blends or modifications to impart a more hydrophilic nature, [34,48,87] with some focusing on forming TFCs for ultrafiltration membranes, where the PES/PSf is the support layer [48]. These polymers are both suitable for the phase inversion [64,89] and electrospinning techniques [34], while we were unable to identify examples of the other techniques mentioned in this review.

An example of some of the methods that researchers employ to improve the hydropathy is provided by Yoon et al. [34]. Here, the authors demonstrated the use of simple methods to acquire what most studies achieve through more complicated composites [69,90,91,92], that is producing a PES membrane with enhanced mechanical strength and improved hydrophilicity. In their study electrospun PES fibers’ mechanical strength was improved by causing fibers to fuse at the joints as a result of different solvent (% N-Methyl-2-pyrrolidone, NMP) blends during the electrospinning process (Figure 8). The fibers were also oxidized to enhance their hydrophilicity. Figure 8 and Figure 9 reveals the effectiveness of this approach.

PES can also be readily used with the phase inversion technique [87,90,91]. Another example where researcher attempt to enhance the hydrophicity of PES is provided by Li et al. [90]. Here, the authors used phase inversion and TiO_2_ nanoparticles to improve both the flux (by 30%) and the strength of the resulting membrane (Figure 10). These results demonstrate that fabrication methods can be modified to improve the performance of membranes and simultaneously provide a method to add further functionality.

### 3.2. Polyacrylonitrile

Polyacrylonitrile (PAN) is another very popular polymer for filtration applications [74]. Like PES, this is due to its desirable physical and chemical properties, particularly its solvent resistance [30,74]. As with PES, PAN is hydrophobic and thus is often blended with more hydrophilic polymers (often chitin or cellulose based polymers) [30,93] to improve its compatibility with aqueous systems. However, PAN is more hydrophilic than PES or PSf and furthermore is more naturally anti-fouling [89], making it a common choice for water filtration membranes [30,50,93,94,95]. A drawback of PAN however is its poor solubility, with the polymer being only soluble in strong polar solvents such as NMP, DMA or DMF. PAN is also suitable for electrospinning or phase inversion techniques.

An example demonstrating a typical use of a PAN based filter membrane is provided by Yeh et al. [96] Here, the authors electrospun PAN as the porous support, in a thin film composite (TFC) filter membrane. Upon this nanofiber support, a thin layer of cellulose nanofiber was cast and lastly up top of that, a graphene oxide layer also deposited. This resulted in a membrane that maintained a high permeate flux (2.2 kg m^2^ h^−1^) while showing excellent ethanol dehydration properties. This study can also be considered an example for the trend of using polymers like PAN and PES in composites with more hydrophilic polymers.

### 3.3. Cellulose and Chitin Derivatives

Biologically derived polymers (such as cellulose or chitin) are obvious choices for aqueous filtration applications, largely due to their innate hydrophilicity and abundance [47,87,97]. However, unlike synthetic polymers, these bio-based polymers often lack the solubility to be easily processed, and for this reason exist as regenerated [98] or derived forms [87,99,100], or are processed without dissolving them, using more esoteric techniques to fabricate composite membranes such as casting methods or spray coating [47,49,97,98]. Thus, these polymers are often blended with other polymers (such as PES or PAN) or used as a barrier layer on more easily processed supports [63,97]. Alternatively, chemically modified derivates, that are more soluble such as cellulose and chitin derivatives (cellulose acetate or chitosan) can be electrospun [95,100,101] or cast using phase inversion [87,99]. Due to is abundance and cost effectiveness, cellulose and its derivatives can be found in many commercially available filters with many filter paper products consisting of cellulose or its variants.

An example of cellulose being processed without dissolving it provided by the work of Wang et al. [97]. Here the authors electrospun PAN to form a fiber mat with a mean fiber diameter of 150 ± 10 nm; after this, a barrier layer of cellulose nanofibers was added using a spray coating method, forming an ultrafiltration thin film composite (TFC) membrane. The spray coating method uses a cellulose nanofiber suspension, thus avoiding the need to dissolve the cellulose. The authors also used PET as a backing for the PAN electrospun fibers, which is a very common practice in preparing such membranes [48,50,102].

### 3.4. Polyvinylalcohol

Polyvinylalcohol (PVA) is a hydrophilic, water soluble polymer, and thus must be covalently crosslinked if it is to be used as the sole component of a filter membrane [28,41,103]. This gives PVA membranes an added optimization parameter, the crosslinking densities. The hydrophilicity makes it an attractive material for membranes, however, due to its poor mechanical properties [74], especially when swollen [104], PVA is often used in composite membranes with another polymer such as PAN [50].

An example of the use of PVA as a barrier layer is provided by Tang et al. [103]. In their study a barrier PVA layer was added by UV crosslinking to electrospun PVA fibers. To prevent deep penetration of the aqueous based barrier layer solution, the crosslinked (via glutaraldehyde) electrospun fibers were soaked in a borax solution which filled the pores. Thus, the barrier layer remained at the fiber mats surface and was crosslinked through UV light. The barrier layer consisted of an aqueous solution of chemically modified PVA with UV crosslinking capability.

### 3.5. Polyvinylidene Fluoride

Polyvinylidene fluoride (PVDF) is a fluorinated hydrophobic polymer with good chemical and thermal stability and unique electric properties [74,89,105,106]. Owing to these properties PVDF is used commercially and industrially in many membranes’ applications, such as water treatment, biomedical filtration (western blotting) and electronic components (such as batteries [105,107]).

Atypically, in this case PVDF’s high hydrophobicity is responsible for its use in the biomedical filtration field as it non-specifically binds amino acids, allowing for protein removal [108] and it is often utilized in water desalination as opposed to simple filtration [109,110].

Membranes prepared from PVDF are often prepared by phase inversion. [14,77] PVDF is unique compared to other membrane polymers in that it has piezoelectric properties and possesses high electrochemical stability [106,111,112] and thus these membranes are often used an electrode separators in batteries [105,106,107] and other electronic applications [105,111,113].

However, simple filtration applications have also been explored [12,25,77,114]. For these simple filtration applications, the focus is mostly on fabricating hydrophilic PVDF membranes, as exampled by Zhao et al. [25]. In this work, amphiphilic triblock polymers were blended with PVDF to produce membranes with hydrophilic surfaces that showed superior anti-fouling properties and significantly better flux rates.

A table detailing the polymer used, the fabrication method and some critical membrane properties is shown in Table 2. Notice the wide range of fluxes and pressures used to test these fluxes, illustrating a lack of convention for the systematic study of membranes. This is attributed to the broad conditions that membranes are used in and the multitude of applications that they are required for.

## 4. Membrane Modification for Antifouling and Specificity

Modification of membrane properties to provide specific functions is a growing trend in filtration membrane research. This is a logical progression of membrane technology, where more custom applications find the need for specialized membranes. This section details some of the targeted areas associated with the development of specialized liquid filtration membranes, in particular, antifouling functionalities and targeted filtration of specific compounds or organisms. These areas dominate the research in modified filtration membranes due to the growing demand of such materials for pollution and disease control.

Membrane modification techniques, such as implementing TFCs, ref. [47,63,96] play a large role in this field, as well as techniques such as atom transfer radical polymerization (ATRP) [7,123,124] or reversible addition−fragmentation chain-transfer (RAFT) [123]. Additionally, incorporation of smart materials such as conducting polymers can allow for control over recognition events [125] or control of the antifouling nature or hydropathy of the resulting membrane [126,127,128,129].

### 4.1. Addition of Antifouling Properties

Methods for the addition of antifouling properties can be broken down into four main categories: surface chemistry modification, barrier layer modification, electrical disruption and physical (mechanical) methods. Surface chemistry modification largely involves the addition of antifouling moieties to the liquid-solid interface of the entire filter [12,15,130]. Barrier layer modification mostly applies to ultrafiltration or RO membranes with the use of TFCs and modifying the barrier layer to incorporate antifouling properties [63,64]. Physical methods may involve the use of a physical barrier, physical movement or abrasion of the membrane surface to reduce fouling [11,54,131,132]. Electrical disruption is a method that uses electrokinetic behavior of materials or causes convection disruption close to the surface of the membrane, preventing fouling [11,133].

For surface chemical modification methods and barrier layer modification, PEG based polymers are the most utilized owing to their well-known anti-fouling and protein rejecting properties [13,127]. Another popular surface modification is the use of Zwitter ionic grafts [14,15]. Both of these polymers have excellent antifouling properties and are easily attached to surfaces. Modification of the surface chemistry for membranes prepared through phase inversion usually involves addition of a compound that aggregates at the surface during the phase inversion process, often a surfactant such as Pluronic F127 [92] but these additives are not limited to surfactants [25,67,134].

A good example of surface chemistry modification is provided by Kolewe et al. [130] where electrospun cellulose acetate mats were functionalized with a Zwitter ion-based polymer, poly(2-methacryloyloxyethylphosphorylcholine) (polyMPC), which was bound to the surface of the fibers by physisorption to a polydopamine layer polymerized to the surface (Figure 11). These fibers demonstrated excellent resistance to fouling by protein and bacteria.

An interesting method for developing antifouling membranes utilizes alternating electric fields [131,132,135]. The principle behind the electrokinetic anti-fouling method is that the alternating electric field causes a turbulent Debye’s double layer that prevents the accumulation of foulants. Furthermore, there is an electro-osmotic force that drives solution flux which further improves transmembrane flux rate [11,131,133]. Li et al. [131] described such a method, with a graphene/polyaniline (PANI) coated polyester filter cloth (Figure 12). In their work Li et al. ascribe the improved flux and antifouling properties to the conductance with a correlation between higher conductance and higher flux and superior anti-fouling properties.

### 4.2. Addition of Selective Filtration Properties

Selective filtration refers to the ability to selectively remove or concentrate a contaminant or desirable component from a source by a filtration process. As an example, the western blotting technique can be considered a protein selective filtration process [136,137], and ion exchange membranes can be considered an ion selective filtration process [138].

Designing a selective filter requires significantly more thought than fabricating a simple filtration membrane, owing to the need for a specific recognition/exclusion element in the filter material. The most common examples of this are proton exchange membranes [139], however these are fairly distinctive when compared to other selective filters. The applications of selective filters are often different than those of the typical liquid filtration applications. From ion exchange/exclusion membranes [4,87,140] in such applications as batteries [107] and other electronics [141], to specific pollutant removal [2,142,143,144,145], to medical applications with biomolecule [143,146] extraction/separation, selective filters are generally used for higher-value applications.

However, water purification still plays a large role in this field of research with membrane modification focusing on the specific removal of bacteria, viruses and pollutants [2,3,6,7,8,118,143,146]. As an example, Ma et al. [118] functionalized the surface of PAN electrospun fibers with imidazole or ethylene glycol containing polymers. The presence of the positively charged imidazole resulted in a greatly enhanced retention of viruses and bacteria, whereas the ethylene glycol based polymer had the opposite effect, with lower bacteria retention, when compared to commercial membranes.

Proton exchange membranes have gained interest due to their part in hydrogen fuel cells and thus due to their part in renewable energy, with Nafion membranes being the industry leader [147]. However, there are a number of other candidates for proton exchange membranes [72,139,148]. PES derivatives are one such polymer that have potential in this area. In a study published by Wang et al. [139], the phase inversion technique was used with sulfonated PES based polymers to produce excellent proton exchange membranes, with high water uptake and ion exchange capacity.

While proton exchange membranes have received more interest due to their part in renewable energy, these are not the only ion exchange/exclusion membranes that are being extensively researched. For instance, Qu et al. [140] developed amine and sulfonic acid functionalized PAN/PEO based membranes using the phase inversion technique. These membranes showed selective rejection between Na_2_SO_4_ and MgCl_2_ depending on the functionalization (Figure 13).

Additionally, membranes for the specific removal of pollutants, nitrates [2,144] and sulphates [3,149] are receiving greater interest due to environmental concerns.

Selective filtration may require or rely on surface functionalization. A method that is potentially underutilized for surface functionalization is the diazonium salt grafting method [150,151,152]. This method involves reducing a diazonium salt to form radical ions that covalently attach to organic surfaces. This allows for a range of functional groups to be added and work has been undertaken to control the process, allowing monolayers comprised of specific groups to be added [153]. Such a technique is an excellent choice when making membranes for selective filtration, such as enhancing cation exchange membranes [138,154]. For instance, in research by Liu et al. [138], grafted polyethyleneimine layers using the diazonium salt grafting method to a commercially available CEM membrane. This added layer benefitted the membrane by forming a more homogenous surface alleviating concentration polarization effects. Furthermore, salt diffusion was suppressed enhancing the current efficiency.

## 5. Conclusions

This review takes a close look at the current research on liquid filtration technology. The polymer materials used, the fabrication techniques and the methods for modifying the membranes for more advanced filtration applications are discussed. The summation of methods in recent literature reveals that there are two methods for fabricating liquid filters that are more commonly used: electrospinning and phase inversion. Additionally, the formation of TFCs is a common method being utilized by researchers. The prevalence of TFCs in recent literature is due to that they provide an effective method to drastically improve existing membranes and allows for better rejections at higher fluxes not otherwise achievable. Looking at the materials chosen for the fabrication of filtration membranes, recent literature demonstrates that the polymer materials chosen are often selected due to their chemical and temperature resistance and then their physical properties but most critically they are chosen based on the fabrication method under investigation or intended to use. It is predicted that providing additional functionality to membranes is likely to be a key trend in future research, with provision of antifouling, and/or selective filtration membranes being important targets.

## Figures and Tables

**Figure 1 polymers-14-04059-f001:**
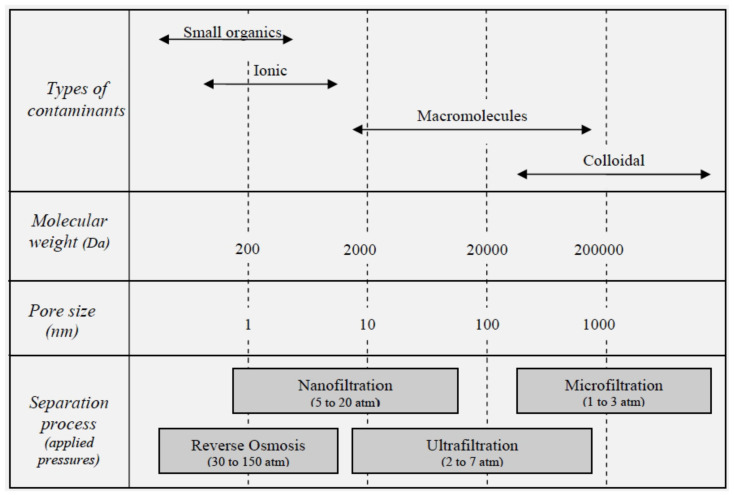
Filtration levels with equivalent particle rejection ranges and molecular weight cut-offs.

**Figure 2 polymers-14-04059-f002:**
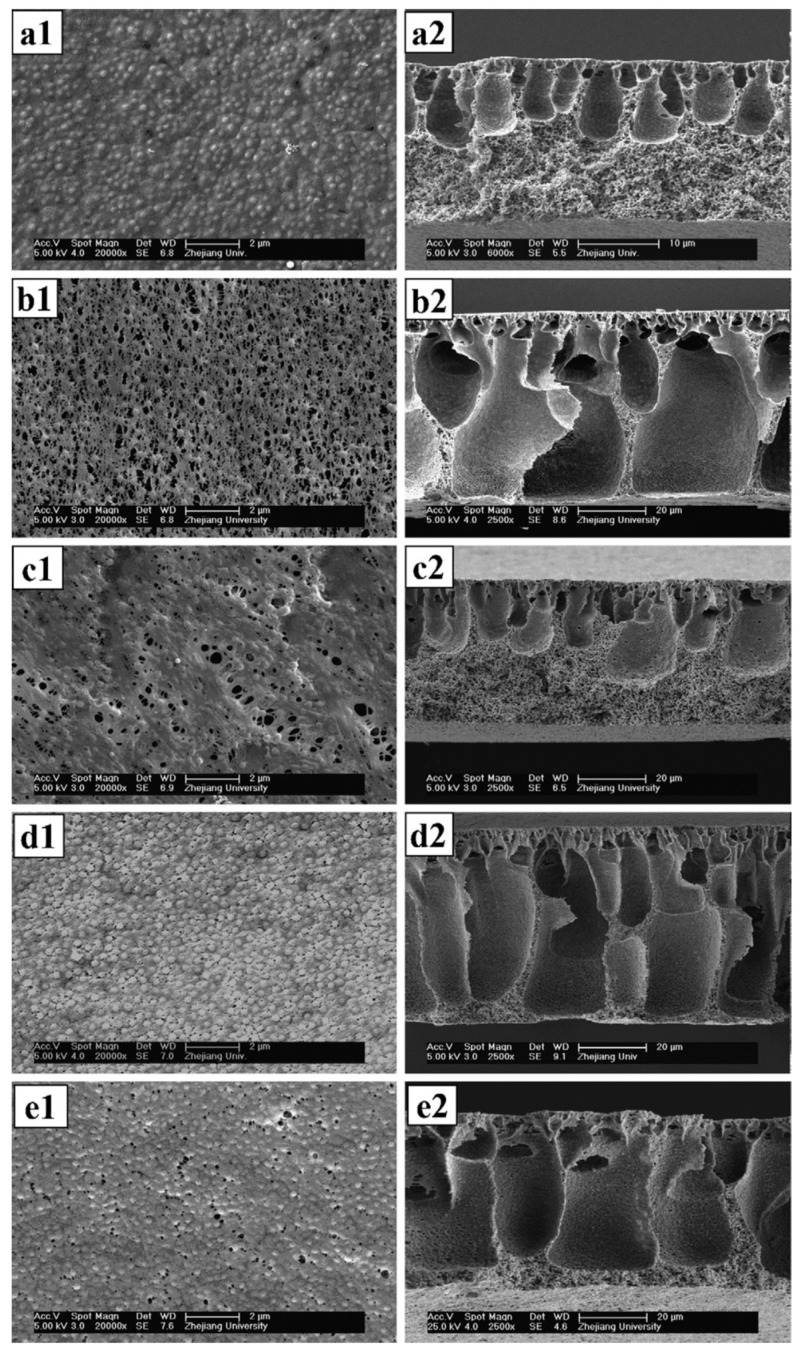
SEM images of top surface (left) and cross-section (right) of the membranes. (**a1**,**a2**) Pure PVDF, (**b1**,**b2**) PVDF/EPTBP, (**c1**,**c2**) PVDF/ACPS, (**d1**,**d2**) PVDF/HPE-g-MPEG and (**e1**,**e2**) PVDF/PEG. Provided with permission from Elsevier [25].

**Figure 3 polymers-14-04059-f003:**
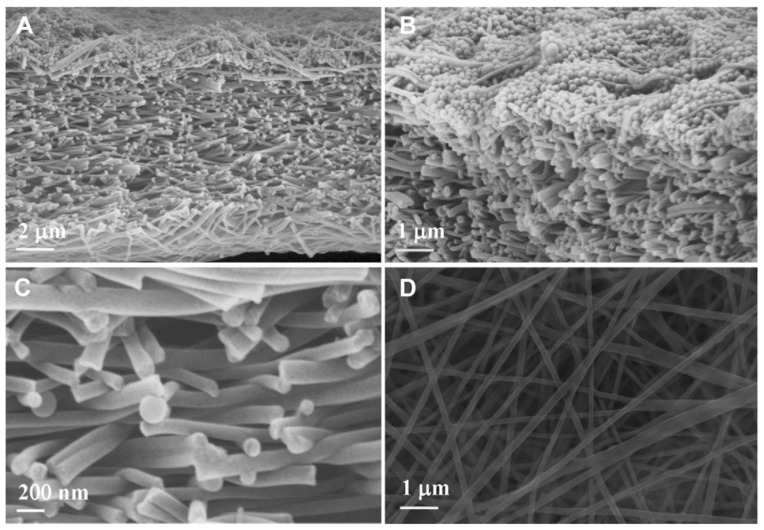
SEM images of the M2 membrane after particle rejection test in different views: (**A**) cross-section; (**B**) top layers of cross-section; (**C**) middle part of cross-section; (**D**) bottom surface. Provided with permission from Elsevier [28].

**Figure 4 polymers-14-04059-f004:**
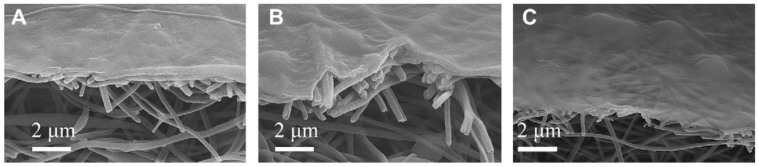
Cross-section SEM images thin film composite membranes of cellulose (**A**), chitin (**B**), and cellulose-chitin blend (**C**) barrier layers prepared by ionic liquid regeneration. Provided with permission from Elsevier [47].

**Figure 5 polymers-14-04059-f005:**
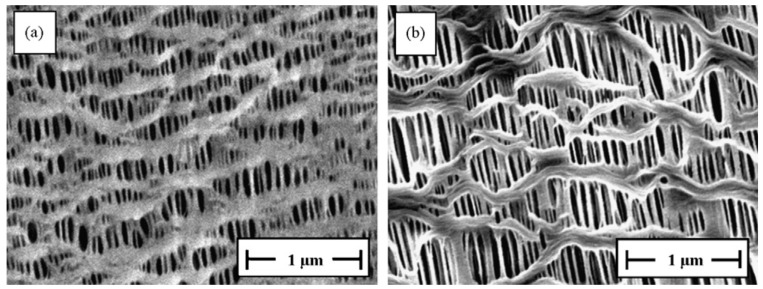
SEM micrographs of the surface of microporous membranes (20 m thick): (**a**) PP and (**b**) HDPE; DR = 90, H-AFR, cold stretching of 55%, followed by hot stretching of75%. Provided with permission by Elsevier. [53].

**Figure 6 polymers-14-04059-f006:**
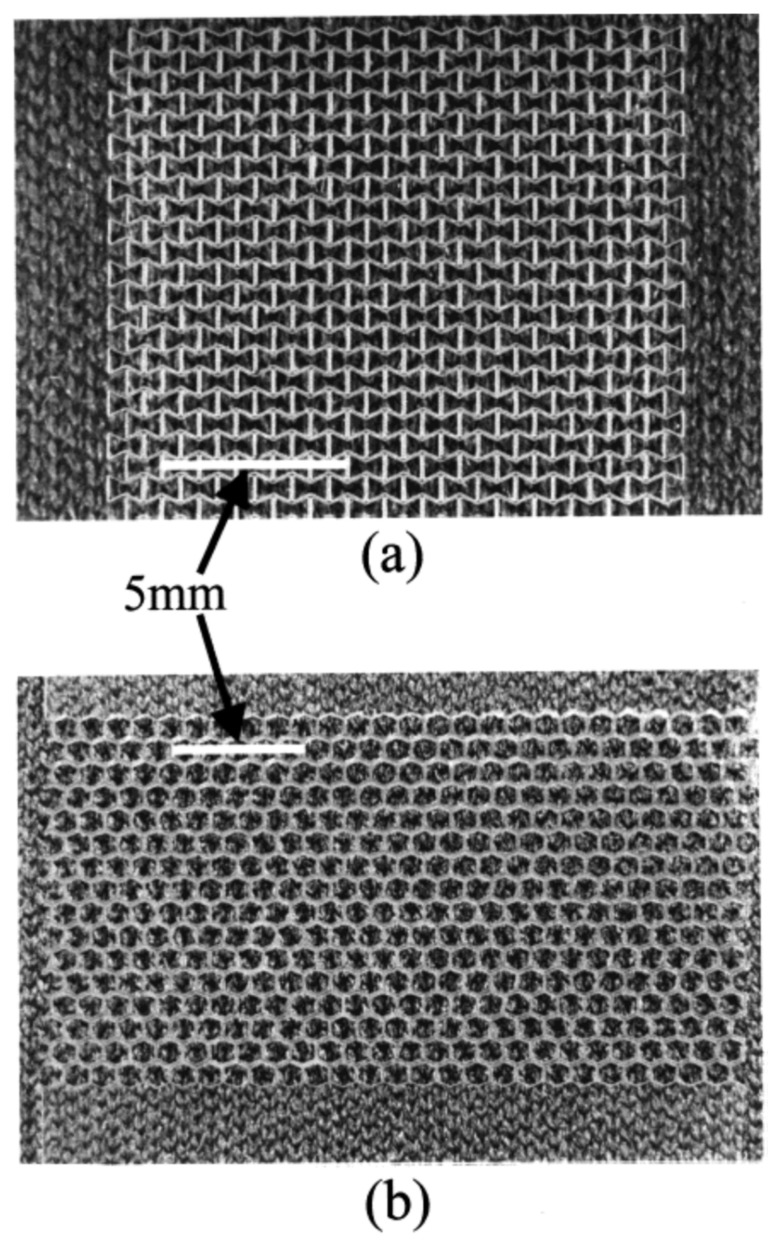
(**a**) Polymeric re-entrant honeycomb membrane; (**b**) polymeric conventional honeycomb membrane. Pores are approximately 1 mm in width (along the x direction). The membranes were fabricated by direct femtosecond laser ablation in air, with pulses at 790 nm (170 fs). Provided with permission from American Chemical Society [55].

**Figure 7 polymers-14-04059-f007:**
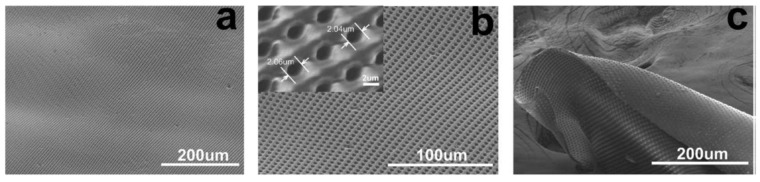
SEM images of (**a**,**b**) the obtained membrane with a well-distributed ordered cylindrical straight through-pore structure (pore size: 2 μm, distance between adjacent pores: 2 μm) in a large area from the imprint process, and (**c**) the bending of the membrane edge for clearly observing the through-pores. Provided with permission from IOPScience [56].

**Figure 8 polymers-14-04059-f008:**
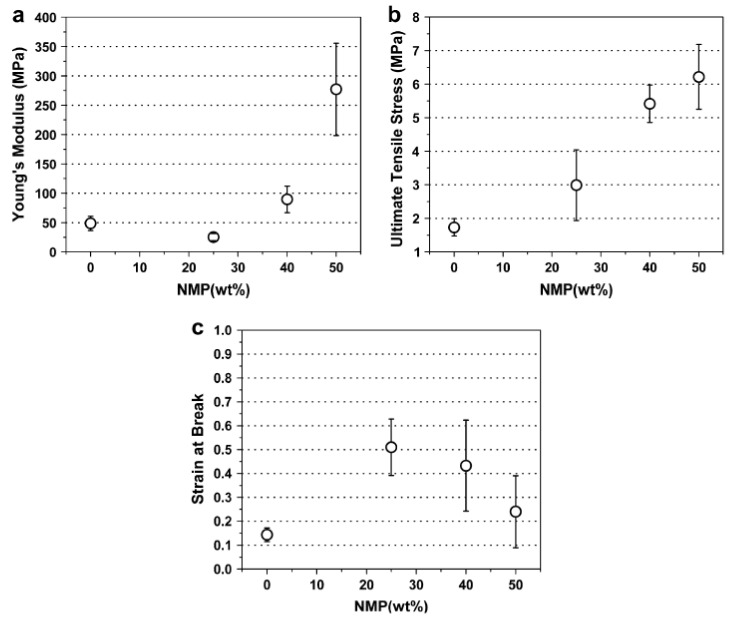
Mechanical properties of electrospun PES membranes as a function of the mixed solvent: (**a**) Young’s modulus, (**b**) ultimate tensile strength, and (**c**) strain at break. Provided with permission from Elsevier [34].

**Figure 9 polymers-14-04059-f009:**
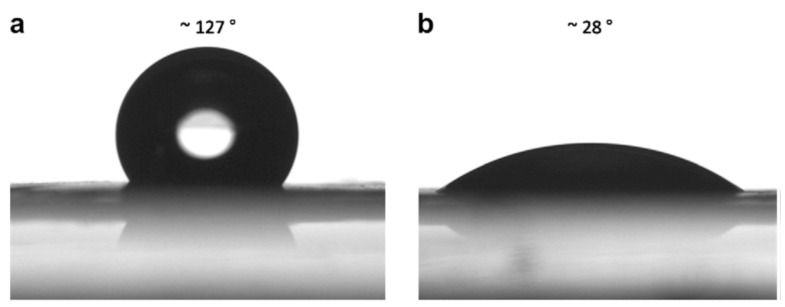
Water contact angles for oxidized (**a**) and untreated (**b**) electrospun PES membranes. Provided with permission from Elsevier [34].

**Figure 10 polymers-14-04059-f010:**
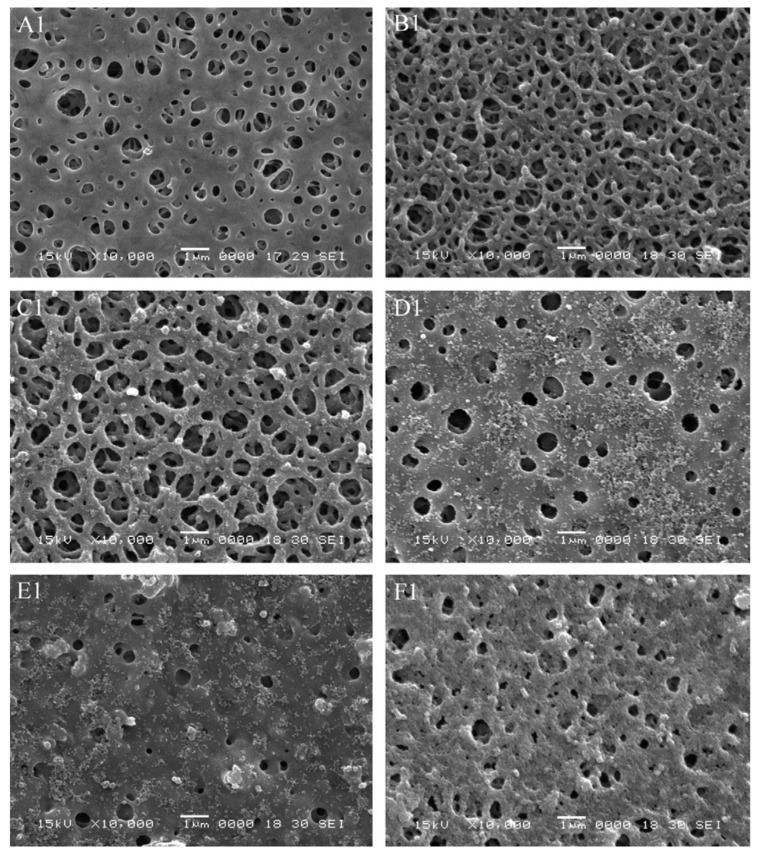
SEM images of the top surface morphology of the membranes with different TiO_2_ content: (**A1**) 0 wt.%, (**B1**) 1 wt.%, (**C1**) 2 wt.%, (**D1**) 3 wt.%, (**E1**) 4 wt.% and (**F1**) 5 wt.%. Provided with permission from Elsevier [90].

**Figure 11 polymers-14-04059-f011:**
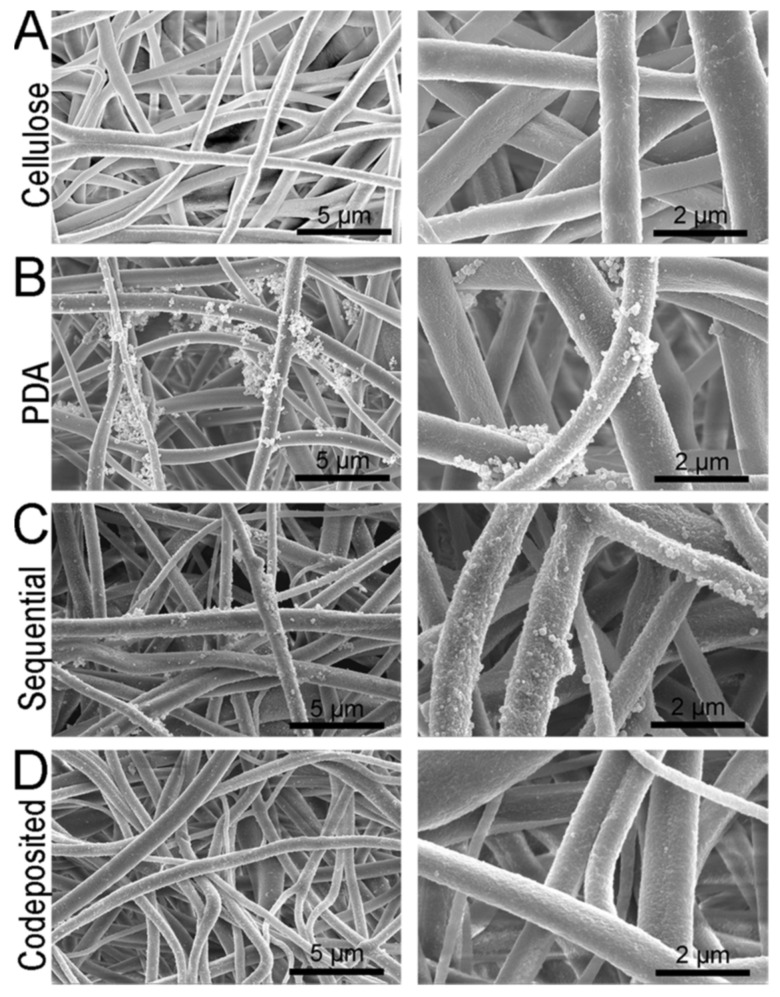
(**A**) SEM micrographs of the cellulose nanofiber mats used as the base materials for this study. The morphologies of (**B**) PDA and (**C**,**D**) polyMPC/PDA (sequential and co-deposited) functionalized nanofiber mats are also displayed. Provided with permission from American Chemical Society [130].

**Figure 12 polymers-14-04059-f012:**
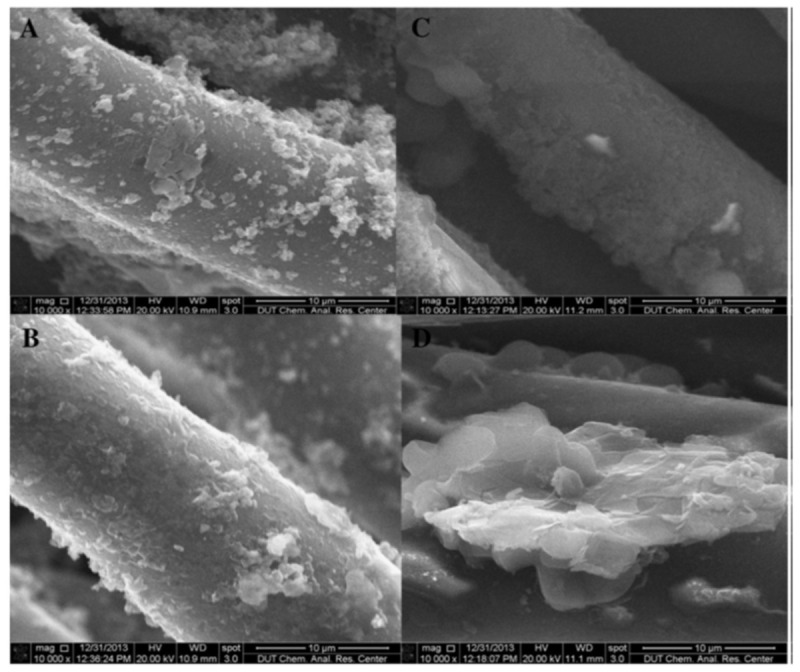
SEM of the modified membranes (**A**,**B**) PANi-PA membrane; (**C**,**D**) Gr/PANi-PA membrane) after use in EMBR. (**A**,**C**) ×10,000, (**B**,**D**) ×10,000. Provided with permission from Elsevier [131].

**Figure 13 polymers-14-04059-f013:**
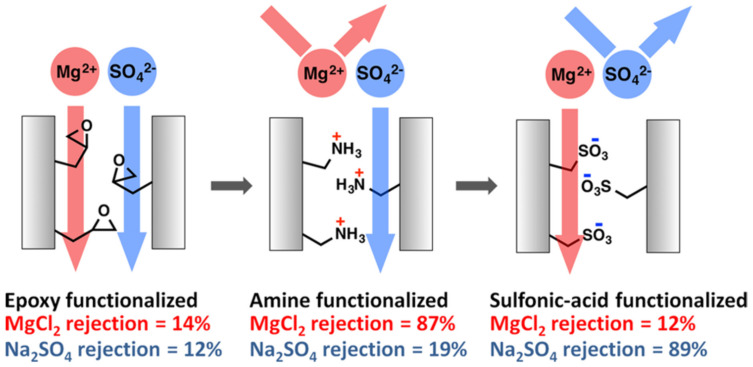
Membrane rejection properties based on surface functionalization. Provided with permission from American Chemical Society [140].

**Table 1 polymers-14-04059-t001:** Physical properties of common polymers used in liquid filtration membranes.

Polymer	Young’s Modulus (MPa)	Tensile Strength (MPa)	T_g_ (°C)	Hydropathy (Philic or Phobic)	Common Solvents	Chemical Resistance
**PES**	25–275 [34]	70–95 [70]	158–228 [71]	Phobic	DMA, DMF, NMP	High [72]
**PAN**	158,000–517,000 [73,74]	7000 [74]	100 [74]	Phobic	DMA, DMF, NMP	Hydrolysis yields copolymers [75,76]
**PVDF**	2.1–8.4 [77]	55,000 [74]	−50–−18 [78]	Phobic	DMA, DMF, NMP	High
**PVA**	6.4–11 [41,79]	50 [79]	76 [80]	Philic	Water, Alcohol	High
**Cellulose**	120,000–220,000 [81]	800–2000 [74]	200–250 [74,82]	Philic	Acetone, DMF	Susceptible to hydrolysis
**Polypropylene**	900–1100 [54]	35 [74]	−25 [83]	Phobic	Naphtha, o-xylene, petroleum ether	High
**Polyamides**	1300–5240 [74,84]	52–83 [74]	42–46 [74,85]	Capable of either	DMA, DMF, NMP	Susceptible to hydrolysis

**Table 2 polymers-14-04059-t002:** Summary of polymers used and the resulting membrane characteristics.

Polymer	Fabrication Method	Filtrant	Filtration Level	Flux (L m^−2^ h^−1^)	Ref
Cellulose Acetate-polysulfone	Phase inversion	Water, protein solution	Ultra	14.1–42 at 414 kPa	[87]
Cellulose Acetate-polyethyleneimine	Phase inversion	Water, protein solution	Micro	6–30 at 69 kPa	[115]
Cellulose and Chitin	Miscellaneous	Water	Ultra	150–450 at 207 kPa	[102]
Polyethersulfone/cellulose acetate butyrate	Dry-jet wet spinning (phase inversion)	BTEX	Not specified	1.45–19.48 at 690 kPa	[91]
Polyethersulfone/Polyamide	Electrospinning and interfacial polymerization	Water	Ultra	12.9–75.1 at 483 kPa	[48]
Polyethersulfone/Cellulose/Polyamide	Electrospinning, casting and interfacial polymerization	water	Nano	5.0 at 210 kPa	[116]
Polyethersulfone/TiO_2_ nanoparticles	Phase Inversion	Water	Micro	3711 at 100 kPa (pressure assumed, but not specified)	[90]
Polyacrylonitrile	Electrospinning	Water	Micro	1.5, pressure not specified	[117]
Polyacrylonitrile	Electrospinning	Water	Micro	712 at 69 kPa	[29]
Polyacrylonitrile with imidazolium cation surface modification	Electrospinning	Water	Micro	989–2185 at 15.86 kPa	[118]
Polyacrylonitrile/chitosan	Electrospinning and cast coating	Water	Ultra	50–65 at 345 kPa	[30]
Polyacrylonitrile/chitosan/graphene oxide	Electrospinning, cast coating and spin coating	Ethanol dehydration	Ultra	2.2 kg m^−2^ h^−1^ at unspecified pressure	[96]
Polyacrylonitrile/poly(ethylene glycol) and cellulose	Electrospinning, casting and photo crosslinking	Water	Ultra	85 at 206.8 kPa	[63]
Polyimide	Phase Inversion	Water	Nano	50 at 1000 kPa	[66]
Polypropylene	Phase Inversion	Desalination	Pore sizes of 10–600 nm	28.92 kg m^−2^ h^−1^ at 3 kPa	[119]
Polypropylene/high density polyethylene	Stretching	Not Specified	19–44 g/m_2_	Not specified	[53]
Polypropylene/graphene oxide	Stretching	Not specified	100 nm to 2 um pore sizes	Not specified	[120]
Poly(styrene-b-lactide)	Self-assembly	Water	24 nm pore sizes	1.15 L m^−2^ h^−1^ bar^−1^.	[59]
Nylon	Electrospinning	Water	Greater than Micro	31 to 593 at 69 kPa	[40]
Poly(vinyl alcohol)	Electrospinning	Water	Micro	11,535 at 17 kPa (Pure water flux)	[28]
Poly(vinyl alcohol)	Electrospinning	Water	Ultra	101.7 at 207 kPa	[103]
PVDF/Synthetic PEG based triblock polymers	Phase Inversion	Water	Micro	Approximately 700 at 100 kPa	[25]
PVDF/HDPE	Melt-processing	Water	Micro	24,000 at 100 kPa	[121]
PVDF	Phase Inversion	Not Specified	Not Specified	Not specified	[77]
Kevlar/PET	Layer by layer assembly	Water	Ultra	1161 to 7585 at 1 kPa	[61]
Linear low-density polyethylene	Imprint and thermal filed induction (Template)	Water	Micro	0.19 at 20 kPa	[56]
PEK-C/PAMAM dendrimers	Phase Inversion and interfacial polymerization	Water and cation separation	Ultra	37.5 to 68.2 at 600 kPa	[122]
Carbonaceous	Te nanowire template	Water	Micro-nano	Not specified	[57]
Hewlett-Packard Color LaserJet Transparency film	Laser ablation	Not specified	Greater than Micro	Not specified	[55]

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
