# Peer review of "Polymeric Materials and Microfabrication Techniques for Liquid Filtration Membranes"

_polymers, 2022, doi:10.3390/polym14194059_

Round 1
Reviewer 1 Report
The manuscript "Polymeric Materials and Microfabrication Techniques for Liquid Filtration Membranes" is well written for a review. there some minor points need to be addressed.
1. The methods how nanofilter membranes can be obtained are very informative. there somehow missing the practical term. Which kind of membranes are often applied in real filters? It would be good for readers to understand which membranes are in general used commercially.
2. Figure 1 gives a nice overview regarding pore sizes and where such is used for but those are not find in samples given in this review. therefore the reviewer suggest adding examples as listed in Figure 1 in the different formation techniques and polymeric materials used.
A Table on the end would achieve such.
3. Some fields such as conducting polymers are missing as materials used in water purification. Are there no industrial or research inventions made of such?
4. Page 15 line 363 (cellulose and chitin derivatives) What are esoteric techniques?
5. In general most those polymer materials are not used in basic forms with fillers such as carbon-particles or other mesoporous materials often used. Those composites are not shown neither discussed in this review. I understand that not all can be shown in a review but then it should be sort of explained in the introduction that only the plain polymers are discussed and presented.
Reviewer 2 Report
Authors of the manuscript titled “Polymeric Materials and Microfabrication Techniques for Liquid Filtration Membranes” have reviewed the methods of fabrication liquid filtration membrane and present main polymeric materials used in this process. Generally, the article is well written and the references to the previous works were adequate selected. Of course, in such a relatively short work it is difficult to present in detail all aspects related to the fabrication and selection of polymeric materials for the production of membranes.
Authors, in contrast to many other review publications in the last decade, have presented the issue of the production of liquid filtration membranes in terms of their further modification and use in specific applications, which distinguishes this work from the others.
I recommend to publish the manuscript after minor linguistic and editorial corrections, e.g. in the list of references, items 1, 68-69, 76 should be corrected or deleted.
